# Seed Availability Does Not Ensure Regeneration in Northern Ecosystems of the Endangered Limber Pine

**Vernon S. Peters * and Darcy R. Visscher** 

Department of Biology, The King's University, 9125 50th Street, Edmonton, AB T6B 2H3, Canada; darcy.visscher@kingsu.ca

**\*** Correspondence: vern.peters@kingsu.ca; Tel.: 1-780-465-3500 (ext. 8127)

**Abstract:** *Research Highlights*: When biotic interactions such as disease alter both the seed production capacity of stands, and seedling survivorship, the relative importance of seed availability versus substrate specificity may alter future regeneration opportunities for plant populations. *Background and Objectives:* We investigated the importance of disease severity, seed availability, and substrate limitation to the regeneration dynamics of the endangered limber pine, *Pinus flexilis*, in two ecosystems with varying forest composition, and different histories of white pine blister rust infection (WPBR; *Cronartium ribicola*). *Materials and Methods:* A total of 17 stands from the montane ecoregion (Alberta, Canada) were sampled for seed production between 2007–2010, seedling density, and age structure. Model selection using an information theoretic approach compared a series of a priori models and their combinations, based on our hypotheses on the role biotic variables play in the regeneration process. *Results:* Despite higher rates of WPBR infection, 2.3 times more seed was available for avian dispersers in the southern ecosystem. Recent seedling regeneration did not correspond to seed production (83 versus 251 seedlings/ha, in southern versus northern ecosystems, respectively), resulting in a seven-fold difference in seed to seedling ratios between ecosystems. Models suggest that disease and vegetation cover were important factors explaining the absence of regeneration in 79.4% of the plots sampled, while basal area (BA) of live limber pine, rocky substrates, ecosystem, South aspects, and slope enhanced limber pine regeneration. Seedling age structures suggest that recent regeneration is less likely in more diseased landscapes, than it was historically (40% versus 72.8% of seedlings < 20 years old, respectively, in southern versus northern ecosystems). *Conclusions:* At the northern limits of limber pine's range, seed availability does not ensure regeneration, suggesting that other environmental or biotic factors hinder regeneration. Regeneration was consistently predicted to be lower in the southern ecosystem than in the northern ecosystem, suggesting that natural regeneration and the potential for population recovery are ecosystem dependent. We recommend that monitoring recent seedlings will aid the identification of biotic and abiotic factors affecting regeneration.

**Keywords:** limber pine; seed production; substrate; regeneration; seedlings; cones; disease; microsites; blister rust; dispersal; nutcrackers

---

## 1. Introduction

One of the fundamental challenges in recovering endangered species is identifying the key factors that allow populations to be self-sustaining despite increased threats. The conservation of plants frequently draws on a rich history of seed ecology studies that test whether species are seed-limited or substrate limited [1]. The tradeoffs between seed size and number of offspring are self-evident in nature with variation ranging 2–3 orders of magnitude in seed size amongst co-occurring species with a similar life form and size at maturity [2]. Seed limitation in coniferous ecosystems may occur

both temporally through intermittent mast years [3,4], seed predation [5], serotiny [6], and the absence of a soil seedbank [7], and spatially through their common reliance on wind for seed dispersal [8]. Substrate limitations to the successful regeneration of small-seeded conifers are well documented in the forestry [9] and fire ecology literature, and much effort has been invested in quantifying both the success and duration of successful regeneration on different substrates [10]. The reproductive ecology of the endangered limber pine (*Pinus flexilis* James), is distinctive in several key ways: it is large seeded, masts, shows some seedbank dormancy, and relies on vertebrate dispersers [7,11], and recruitment may last centuries because key substrates may remain available [12]. When biotic interactions such as disease alter both the seed production capacity of stands, and survivorship across substrates, the relative importance of seed availability versus substrate importance may decline, altering future regeneration opportunities.

Limber pine is a long-lived pioneer species that occupies montane and subalpine habitats [13] throughout the Rocky Mountains, Great Basin, and Sierra Nevadas, extending from Alberta, Canada to California, USA [14]. It has declined range-wide since the introduction of the fungus *Cronartium ribicola* Fisch, which causes white pine blister rust (WPBR), with the most severe infestations in the northern US and southern Canada, concentrated around the international parks of Glacier, MN, USA, and Waterton, AB, Canada [15]. Limber pine was listed provincially as endangered in Alberta in 2008, due to the high mortality caused by WPBR, mountain pine beetle (*Dendroctonus ponderosae*), Hopkins, and limited regeneration opportunities [16]. WPBR monitoring efforts in southern Alberta show an increase in mean limber pine mortality (>1.3 m tall) from 40% to 52% from 1996 to 2009, while in northern Alberta, mean mortality has remained similar from 2003/4 to 2009 (12% to 13%), while infection levels have increased from 2% to 11% [17]. Overall, greater rates of adult mortality than regeneration occur [18]. In Wyoming and Colorado, mean mortality rates of 5.4% were observed amongst limber pine (>1.37 m tall), but could not be attributed to WPBR at the southernmost locations [19]. Lower rates of mortality and infection at the periphery of WPBR limber pine distribution [17,19] suggest that natural seed production and regeneration processes maintain current populations at these locations.

The natural regeneration process of limber pine is reliant chiefly on the mutualistic dispersal and caching of seeds by the Clark's nutcracker (*Nucifraga columbiana* Wilson) in favorable microsites [7]. Nutcrackers are thought to preferentially cache in open sites, including on burns [20], facilitating long-distance dispersal from parent trees and immediate post-fire colonization. Nutcrackers cache in a wide array of sites, including newly disturbed openings [21,22], and have been observed caching at distances up to 33 km from a seed harvest stand [22]. The preferred germination substrates of limber pine are not well documented. Limber pine is able to establish itself in exposed, harsh sites by virtue of its high seedling tolerance to heat and drought [14,23]; however, natural regeneration and survivorship of planted seedlings is higher with adjacent nurse objects like rocks or stumps [24,25]. The timing and abundance of regeneration are variable amongst burns [25]; however, most stand reconstruction studies show evidence of continuous recruitment over decades to centuries [12,26]. Regeneration delays and low stocking rates have typically been interpreted as evidence of seed limitation, caused by inadequate nutcracker behavior, despite no attempt to quantify substrate availability, or seed to germinant ratios [26,27]. Continuous regeneration in 100–600-year-old stands have reported the highest stem densities amongst seedling and sapling age classes, and no evidence of differential survival amongst initial versus later recruits [12], suggesting that the regeneration niche remains open indefinitely by virtue of the open nature of limber pine stands, and continued availability of mineral soil and scree [12].

Reduced seed availability due to WPBR-induced adult mortality may compromise the seed dispersal mutualism with the Clark's nutcracker [28], leading to seed limitation in landscapes with severe WPBR infestation. While several studies report natural regeneration dynamics without addressing the effects of WPBR [12,26], these studies can be viewed as documenting the potential for "in situ" regeneration, which is of critical importance to population maintenance in disease altered landscapes. Regeneration traits of limber pine make it particularly susceptible to seed limitation, notably its early succession status and occupancy of arid habitats [29], large seed, and dependency on vertebrates for dispersal [30].

We investigated the importance of seed versus substrate limitation to regeneration dynamics of the endangered limber pine in two ecosystems with different histories of WPBR infection. One ecosystem, occurring in Kootenay Plains, AB, represents the northern-most populations of limber pine in North America [16,18], and thus is ecologically significant for the study of demographic characteristics. Our objective was to determine whether seedling densities and age structure differ between ecosystems with different WPBR infection levels, and more broadly whether seed limitation and substrate availability remain useful constructs for maintaining populations of endangered trees in disease altered ecosystems. We hypothesize that severe WPBR landscapes have less regeneration than low WPBR landscapes at the northern periphery due to seed limitation, and greater WPBR induced mortality of seedlings over time. We further hypothesize that seed and substrate limitation processes diminish in importance for maintaining regeneration when other ecological processes (i.e., disease) contribute to species decline. These hypotheses have important implications for recovering an early succession species like limber pine, since fire is presumed to be critical for creating new microsites for regeneration, and substrate controls to regeneration in existing stands have not been investigated. Disease may compromise the importance of seed availability (directly) and substrate availability and competition (indirectly) to the regeneration process by reducing seed production and increasing seedling mortality on favorable microsites.

## 2. Materials and Methods

### 2.1. Study Area

Our study was conducted at the northern limits of limber pine range in the Rocky Mountains of North America. Suitable habitat occurs in the montane region, where the climate is primarily Cordilleran, and has a mean daily temperature of 13 °C from June to August, the warmest months, and −10 °C from December to February, the coldest months. A July precipitation maxima occurs and 580 mm precipitation occur annually [31]. We selected two distinct geographical regions of the Montane Ecoregion in Alberta, Canada, that were approximately 400 km apart. Our northern study area (latitude 52.00° N, longitude 116.50° W), henceforth referred to as the "northern ecosystem", is situated within the North Saskatchewan River Valley, extending from the Kootenay Plains Ecological Reserve to provincial forests adjacent to Abraham Lake. Our southern study area (latitude 49.60° N, longitude 114.20° W), henceforth referred to as the "southern ecosystem", extended east of the Rockies to include the Porcupine Hills formation, and foothills ridges in the Crowsnest Pass (Figure 1). Climate records from 1999 to 2008 indicate that the northern ecosystem was cooler and drier over this period (mean low, −4.5, mean high 7.2, 469 mm precipitation, 13.09 MJ/m$^2$ estimated incoming radiation) than the southern ecosystem (mean low, −1.4, mean high 10.4, 525 mm precipitation, 13.65 MJ/m$^2$ estimated incoming radiation) [32].

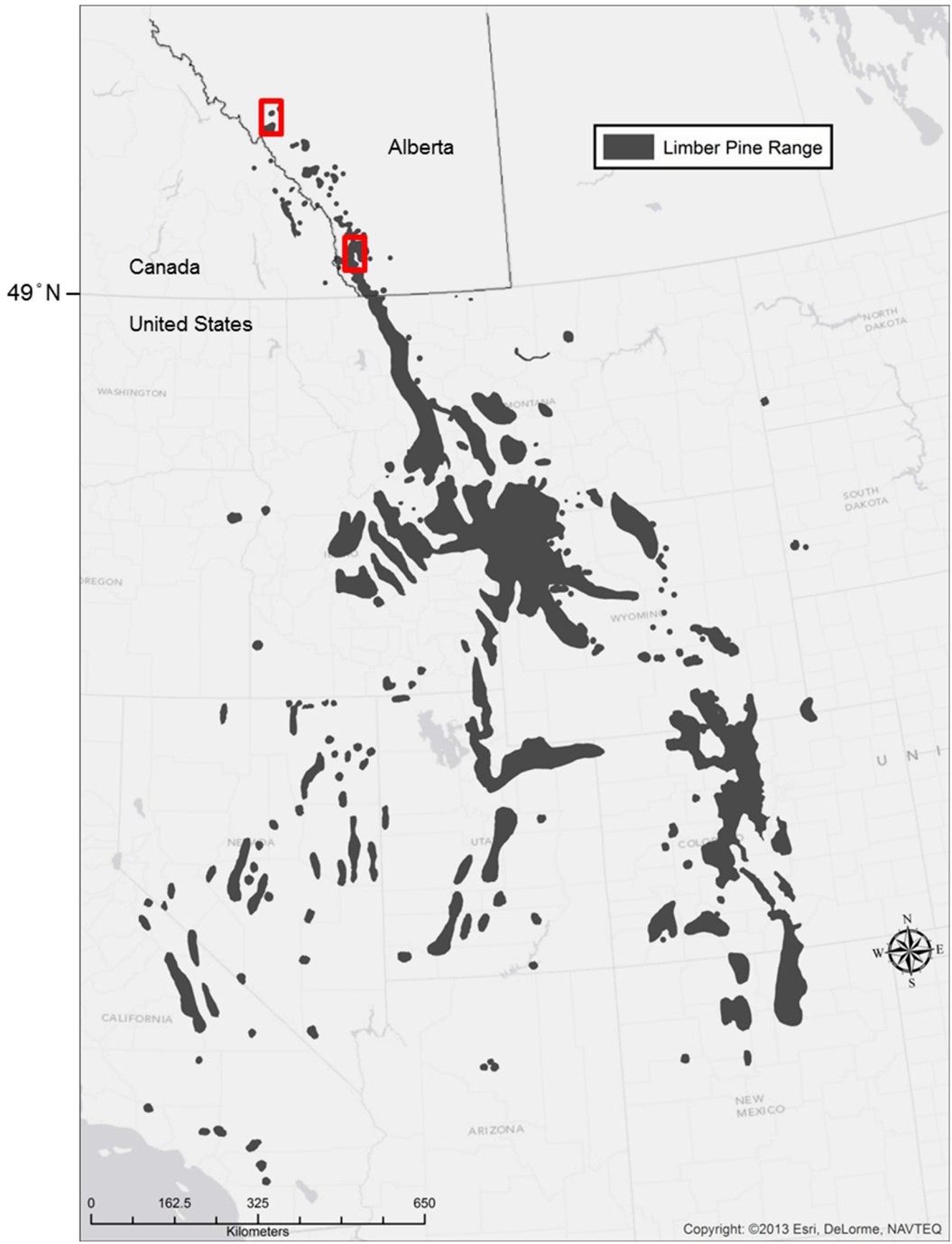

**Figure 1.** Map showing the northern and southern limber pine study ecosystems (rectangles) in Alberta relative to limber pine distribution in North America (reprinted with permission from [33]).

Limber pine in the North Saskatchewan River Valley (northern ecosystem), is found on a variety of aspects (116°–270°) on rocky ridges, scree slopes, planes, and erosional cliff slopes with alluvial deposits, at elevations of 1348–1455 m. There, limber pine occurred in pure stands, or in association with white spruce, *Picea glauca* (Moench) Voss, or lodgepole pine, *Pinus contorta* Douglas Loudon. In the southern ecosystem, limber pine occurred on southwest to west-facing (214°–270°) on the tops of rocky ridges, at similar elevations of 1300–1504 m. There, stands occurred either as limber pine or limber pine/Douglas fir (*Pseudotsuga menziesii* (Mirb) Franco) mixtures.

The historic fire regime for Alberta's limber pine is uncertain, although as a montane species, it may typically experience a mixed severity fire regime [34]. This regime may have differed between

the two study regions, with less frequent crown fires predominating in the northern study sites where adjacent subalpine fuel complexes are present, while more frequent, lower intensity fires from adjacent grasslands likely influenced southern study areas. Wildfires have been actively suppressed in both regions for most of the 20th century, and cattle grazing has replaced wildfire in the southern study area. White pine blister rust was detected in 1952 in our southern study area [35], where in 2009 it infected 38% of our study trees (*n* = 3 study stands; a subsample of stands from Smith [17]). Surveys in our northern study area documented both the occurrence and rapid increase in WPBR infection (1% live tree infestation in 2003, 20% in 2009; *n* = 3 study stands sampled from Smith [17]).

*2.2. Sampling Methods*

We selected stands based on the presence of prior WPBR infection monitoring data, and their representation of tree species combinations and site attributes associated with limber pine populations (Table 1). Detailed WPBR infection and mortality data was available for three stands from each study area (see, [17]). Study stands were situated from 300 m to 10 km apart from the nearest stand, and >5 ha in size. Marked changes in site types and associated changes in tree species composition frequently occurred in proximal stands. We sampled nine stands in the northern study area, which ranged from pure limber pine, to associations with white spruce, or lodgepole pine. In the southern ecosystem, we sampled eight stands, containing either limber pine, or limber pine and Douglas fir associations.

Stands were sampled for regeneration during the summers of 2008–2009. From a randomly chosen point at the forest edge, we commenced sampling 50 m into the forest. We sampled four sites in each stand along a transect, at random distances of 80–120 m from adjacent sites, due to intervening gaps in limber pine distribution. Each site had a minimum of 10 proximal trees of cone-bearing size. Live and dead BA area by tree species, site-level environmental variables, and annual cone availability for regeneration (2008–2010) from 10 mature trees at each site (40 trees per stand) were recorded at these sites, as previously described [5]. At each of the four sample sites within each stand, three 25 m$^2$ regeneration plots were situated 15 m from the transect (chosen randomly from the four cardinal directions; *n* = 12 plots per stand, totaling 300 m$^2$). Thus, 108 and 96 plots were sampled in northern and southern study areas, respectively. Plots were searched systematically for all regeneration ranging from current year germinants to seedlings < 50 cm in size. The height and age of all seedlings were recorded. Seedling age was estimated non-destructively by counting all visible terminal bud scars from the leader tip down to ground level. On all seedlings, terminal bud scars were readily visible for recent years, but could not be reliably counted beyond 20 years for many seedlings; consequently, 20 served as the cut-off value to distinguish between "recent seedlings" and older seedlings.

In order to quantify substrate and seedling regeneration relationships amongst stands and study areas, a 4 m$^2$ substrate plot was nested within the 25 m$^2$ plot. The larger plots were oriented to maintain homogeneity of substrate conditions throughout. Substrate abundance was recorded according to the following % cover classes: 1 = 0–5%, 2 = 5.1%–10%; 3 = 10.1%–25%; 4 = 25.1%–50%; 5 = 50.1%–75%, and 6 = 75.1%–100%. During data analysis, we substituted the categorical values with the median value from each cover class range. Substrates were classified as rock, scree (rock < 10 cm diameter), mineral soil, humus, needles, leaf litter, and moss. Additionally, we recorded the actual rooting substrate of each seedling according to the listed substrate categories. Recent litter and humus at the base of each tree were carefully brushed aside. Due to the frequent longevity of seedlings on substrates, many seedlings were assigned two or more rooting substrates (i.e., mineral and needle). Seedlings were classified as healthy (uninfected with WPBR), infected with inactive branch canker, or active branch canker. To evaluate the effects of vegetation cover on seedling abundance, we independently recorded grass cover, herbaceous cover, ground shrub cover (0–0.15 m) and low shrub cover (0.16–1 m), and then summed values to construct an ordinal leaf area index based on these four strata for vegetation up to 1m in height, within a 4 m$^2$ plot. The percent cover of known WPBR hosts (to genera) was recorded separately as well (i.e., *Ribes* and *Castelleja).*

**Table 1.** Stand structure data and site attributes for study sites used to evaluate recent limber pine regeneration relative to seed availability and substrate conditions. Live trees were tallied separately from standing dead trees in basal area counts (BA).

| Stand | Live Stand Composition | | | | | | Mortality | | | | | |
|---|---|---|---|---|---|---|---|---|---|---|---|---|
| | *P. flex* BA (m²/ha) | *P. cont* | *P. glau* | *P. menz* | *P. trem* | *P. flex* Relative Abundance (% Total BA) | *P. flex* (m²/ha) | *P. flex* Relative Abundance (% Total BA) | Latitude° | Longitude° | Aspect° | Slope° |
| Northern Ecosystem | | | | | | | | | | | | |
| KP1 | 8.3 | 0.2 | 0.8 | 0.0 | 0.0 | 89.3 | 1.2 | 100.0 | 52.005 | −116.466 | 124 | 15.0 |
| KP2 | 4.7 | 0.3 | 3.5 | 0.0 | 0.0 | 54.9 | 0.0 | 0.0 | 52.002 | −116.466 | 135 | 5.0 |
| KP3 | 3.8 | 0.0 | 2.5 | 0.0 | 0.0 | 60.5 | 0.2 | 54.5 | 52.046 | −116.397 | 270 | 22.0 |
| KP4 | 2.7 | 3.8 | 0.2 | 0.0 | 0.0 | 40.0 | 0.3 | 100.0 | 52.054 | −116.390 | 200 | 13.0 |
| KP4B | 5.4 | 0.0 | 0.0 | 0.0 | 0.0 | 100.0 | 4.2 | 100.0 | 52.049 | −116.388 | 221 | 35.0 |
| KP5 | 1.7 | 3.0 | 0.2 | 0.0 | 0.0 | 34.5 | 0.0 | 0.0 | 52.247 | −116.435 | 146 | 28.0 |
| KP6 | 4.5 | 0.0 | 1.2 | 0.0 | 0.0 | 79.4 | 0.0 | 0.0 | 52.255 | −116.395 | 124 | 26.0 |
| KP7 | 5.3 | 0.0 | 2.3 | 0.0 | 2.5 | 52.5 | 0.0 | 0.0 | 52.271 | −116.385 | 169 | 36 |
| KP8 | 6.0 | 3.5 | 0.0 | 0.0 | 0.0 | 63.2 | 1.2 | 100.0 | 52.007 | −116.460 | 116 | 23.0 |
| Southern Ecosystem | | | | | | | | | | | | |
| Charolais | 5.7 | 0.0 | 0.0 | 0.7 | 0.0 | 89.5 | 3.0 | 100.0 | 49.678 | −114.008 | 259 | 33.0 |
| Calvin | 8.2 | 0.0 | 0.0 | 4.0 | 0.0 | 67.1 | 0.7 | 100.0 | 49.858 | −114.240 | 270 | 21.5 |
| Lundbreck | 7.3 | 0.0 | 0.0 | 0.7 | 0.0 | 91.7 | 2.2 | 67.7 | 49.584 | −114.189 | 214 | 20.0 |
| East Sharples | 5.0 | 0.0 | 0.0 | 6.2 | 1.7 | 39.0 | 0.5 | 71.4 | 49.916 | −114.993 | 259 | 10.0 |
| Heath Creek | 5.5 | 0.0 | 0.0 | 0.3 | 0.0 | 94.3 | 0.8 | 100.0 | 49.786 | −114.059 | 270 | 38.0 |
| Ross | 5.7 | 0.0 | 0.0 | 2.2 | 0.0 | 72.3 | 3.0 | 90.9 | 49.594 | −114.233 | 243 | 14.0 |
| Welsch | 2.5 | 0.0 | 0.0 | 1.2 | 0.0 | 68.2 | 4.2 | 100.0 | 49.681 | −114.953 | 260 | 22.0 |
| Simps | 11.2 | 0.0 | 0.0 | 0.0 | 0.0 | 100.0 | 3.7 | 100.0 | 49.667 | −114.783 | 265 | 23.0 |

Notes: Species abbreviations are: *P. flex* = *Pinus flexilis*, *P. cont* = *Pinus contorta*, *P. glau* = *Picea glauca*, *P. menz* = *Pseutotsuga menzesii*, *P. trem* = *Populus tremuloides*. Mortality data includes all causes of death, and is shown solely for *P. flex* which accounted for the majority of standing dead stems in most stands; [a] *P. glau* accounted for 0.50 and 0.17 m²/ha dead BA, in Kootenay Plains (KP2 and KP3 respectively); *P. menz* accounted for 1.03, 0.17, and 0.33 m²/ha dead BA, in Lundbreck, E. Sharples, and Ross, respectively.

*2.3. Statistical Analysis*

2.3.1. Model Identification

To test which ecological factors are associated with limber pine seedling regeneration, we used a generalized linear model framework to fit models to our seedling counts from 25m² sample plots. Additionally, to account for the hierarchical nature of our sampling design, we used a mixed model approach and nested the random effect of plots within sites and stands via a random intercept in all models [36]. We tested whether a zero-inflated modelling approach was required [37]. Model covariates were only retained when correlations of less than 0.5 were measured (9 of 15 covariates retained; Table 2). The descriptions of model covariates are provided in Table 2. Model selection was conducted using Akaike's Information Criterion corrected for small sample size (AICc) to determine which model was the most parsimonious from our a priori candidate model set [38,39]. While care should be taken using a coefficient of determination estimated for general linear mixed models, we assessed model fit using formulations by Nakagawa and Schielzeth [40] and Johnson [41]. Following the identification of the best candidate model, we inspected model residuals to confirm that errors were homogeneous and independent with respect to model covariates.

**Table 2.** A priori hypotheses (models) and their respective covariates and covariate descriptions.

| Hypothesis | Reason | Covariates | Description |
|---|---|---|---|
| Ecosystem | Climate, geology, and forest composition differs with latitude | ecosystem | Factor for northern or southern ecosystem |
| Microsite | Soil moisture and solar insolation vary with aspect and slope | direction slope | Ordinal factor for cardinal aspect (W = 225–314°, N = 315–44°, E = 45–134°, S = 135–224°) [†] Arcsine transformed % slope |
| Seed limitation | Seed availability may control regeneration | cone | Number of cones produced in 2010 mast year |
| Substrate limitation | Substrate availability may limit germination and survivorship | BA_lp mineral rock | Basal area of limber pine Mineral soil cover Rock cover |
| Competition | Seedlings require open conditions | vegetation | Vegetation cover index |
| Disease | Seedlings are susceptible to mortality | BA_dead lp | Basal area of dead limber pine |

[†] South facing aspects were identified as the highest ordinal category based on prior literature regarding microsite preferences.

We analyzed the number of regenerated limber pine seedlings using a two-step procedure. First, we compared four competing model forms using a standardized model with respect to independent variables. Using AICc, we compared Poisson, zero-inflated Poisson, negative binomial, and zero-inflated negative binomial model forms. Secondly, using the model form identified in the previous step, we compared a series of a priori models and their combinations, based on our hypotheses on the role variables play in limber pine regeneration. Our candidate set of models were constructed around the two part nature of the zero inflated model, which contains covariates for both the zero inflated and count component of the model. We included in every model a "null" within the count component related to physical aspects of the sites measured, including ecosystem, slope, and aspect, since there is ample evidence from plant studies that these covariates affect regeneration [1]. Additional to the count component, we added variables related to the hypotheses related to seed and substrate limitation, while the zero component of the model was constructed using variables related to the hypotheses of disease and competition. Our a priori candidate set therefore contained all combinations of the four hypotheses of interest within their respective components of the model. We identified three additional models that include seed limitation, disease, and competition as part of both the count and zero inflated component of the model. All analyses were conducted using the glmmTMB package in R, model selection used bbmle, and ggeffects was used to make model predictions [42–48].

**Table 3.** Model selection results to determine the appropriate model form for the regeneration of limber pine seedlings.

| Model Form | K | LL | AICc | ΔAICc | $W_i$ |
|---|---|---|---|---|---|
| ZIP | 17 | −135.8 | 308.9 | 0.0 | 0.737 |
| ZNB | 18 | −135.6 | 310.9 | 2.1 | 0.258 |
| Negative Binomial | 15 | −142.5 | 317.6 | 8.7 | 0.009 |
| Poisson | 14 | −155.3 | 340.9 | 32.0 | <0.001 |

In all cases we fit the conditional model based on the global model (all covariates listed in Table 2) and included a nested random effects (plots measured at sites in stands) via a random intercept. The zero-inflated portion of the model included only the covariates of basal area of dead limber pine and vegetation cover. Models are shown in decreasing rank, with values for model log-likelihood (LL), number of estimated parameters (K), Akaike's Information Criterion for small samples (AICc), AICc difference (ΔAICc), and model weight ($W_i$).

### 2.3.2. Age Structure Analysis

We used a one-sample, Kolmogorov-Smirnov goodness of fit test (KS-test) for discrete data [49] to test whether seedling ages in each study region reflected a uniform establishment pattern over the past twenty years ($n$ = 68, and 20, in northern and southern ecosystems, respectively). A uniform age distribution, parameterized from 1 to 20, for the full twenty-year period of bud scar ages was used. Additionally, we used a two-sample KS-test for discrete data [49] to determine whether seedling regeneration in the past 20+ years differed between northern and southern ecosystems. Seedlings greater than 20 years were grouped into a single age category for each site. To minimize the likelihood of older seedlings unduly influencing the age distribution tests between study areas, we performed another KS test where all seedlings were grouped into 5 year age categories, and a 20+ year-old category.

## 3. Results

### 3.1. Modeling Results of Regeneration Density

Recent seedling regeneration (<50 cm tall) was 3.02 times greater in the northern ecosystem than in the southern ecosystem (251 vs. 83 seedling/ha; Figure 2). Recent seedlings (<20 years old) were found in all nine stands in the northern ecosystem (67–533 seedlings/ha), while no seedlings were found in two of the eight stands surveyed in the southern ecosystem (0–233 seedlings/ha).

We recorded 0 seedlings at 79.4% of the plots we sampled and initial model selection suggested that a zero-inflated Poisson was more suitable for the data ($w_i$ = 0.737) relative to models without zero-inflation or using the negative binomial distribution (Table 3). Using the zero-inflated Poisson model, we found evidence that a simplified variant of the global model ("final model") was the most parsimonious model ($w_i$ = 0. 564, Table 4) and contained at least one covariate from each of the a priori hypotheses (Table 5). Model selection using corrected Akaike's information criterion (AICc) suggests that there is strong support for the final model, which included seed, substrate, disease, and competition, over any less inclusive model containing three or fewer covariates (ΔAICc ≥ 3.4; Table 4). In the final model, the zero-inflated model component was comprised of covariates related to the disease and competition hypotheses, while the conditional (count) model component was comprised of covariates related to the hypotheses of seed availability and substrate. Aspect, slope, and ecosystem from the null model also contributed to the count component. There was no evidence that the inclusion of seed limitation, disease, and competition covariates into both the count and zero inflated component of the model was warranted (all $w_i$ < 0.05; Table 4) and the covariates were not significant within the models to which they were included (all $p$ < 0.5). The final model adequately described the data, with the marginal model (fixed effects) explaining 68% of the variation in the data, while the conditional model (fixed and random effects) explained 73% of the variation in the data.

**Table 4.** Model selection results from candidate models for the regeneration of limber pine seedlings.

| Model/Hypothesis | K | LL | AICc | ΔAICc | $W_i$ |
|---|---|---|---|---|---|
| Simplified Su+Sd\|D+C [†] | 13 | −136.8 | 301.6 | 0.0 | 0.564 |
| Su+Sd\|D+C | 15 | −135.9 | 304.4 | 2.9 | 0.136 |
| Su+Sd\|D | 14 | −137.4 | 304.9 | 3.4 | 0.104 |
| Su+Sd\|C | 14 | −138.0 | 306.2 | 4.6 | 0.056 |
| Su+Sd+D\|D+C | 16 | −135.8 | 306.6 | 5.0 | 0.046 |
| Su+Sd+C\|D+C | 16 | −135.9 | 306.7 | 5.2 | 0.043 |
| Su+Sd\|D+C+Sd | 16 | −135.9 | 306.7 | 5.2 | 0.042 |
| Sd\|D+C | 13 | −141.3 | 310.5 | 9.0 | 0.006 |
| Sd\|C | 12 | −143.6 | 312.8 | 11.2 | 0.002 |
| Sd\|D | 12 | −144.2 | 313.9 | 12.4 | 0.001 |
| null\|D+C | 11 | −148.5 | 320.4 | 18.9 | <0.001 |
| Su\|D+C | 13 | −146.6 | 321.2 | 19.7 | <0.001 |
| Su\|C | 12 | −147.8 | 321.3 | 19.7 | <0.001 |
| Su\|D | 12 | −148.6 | 322.9 | 21.3 | <0.001 |

[†] Henceforth referred to as the "final model". In all cases the models are zero-inflated Poisson models (with a log link) with nested random effects (plots in sites in stands) via a random intercept in the conditional model. Models are shown in decreasing rank, with values for model log-likelihood (LL), number of estimated parameters (K), Akaike's Information Criterion for small samples (AICc), AICc difference (ΔAICc), and model weight ($W_i$). Note that the null model which contained the variables of slope and aspect as well as the factor of ecosystem was included in all other models. Models are named based on the hypotheses they contain and take the format of count component | zero component, thus the model Su+Sd|D+C contains the variables for the substrate (Su) and seed (Sd) limitation hypotheses in the count component and the variables related to the disease (D) and competition (C) hypotheses in the zero component of the model.

**Table 5.** Covariates included in the final model from Table 4.

| Model Part | Effects | Group | Variance | Std. Dev. | |
|---|---|---|---|---|---|
| Conditional | Random | site:stand | 0.110 | 0.331 | |
| | | stand | $3.15 \times 10^{-9}$ | $5.62 \times 10^{-5}$ | |
| | | **Covariate** | **Coefficient** | **SE** | **P** |
| Conditional | Fixed | Intercept | −2.940 | 1.201 | 0.014 |
| | | Ecosystem (South) | −2.191 | 0.529 | <0.001 |
| | | Aspect 2 | 0.042 | 1.167 | 0.972 |
| | | Aspect 3 | 1.090 | 1.127 | 0.333 |
| | | Aspect 4 | 2.349 | 1.133 | 0.038 |
| | | Slope | 1.366 | 0.910 | 0.133 |
| | | BA limber pine | 0.464 | 0.096 | <0.001 |
| | | Rock | 0.024 | 0.007 | <0.001 |
| Zero-inflated | | Intercept | −1.435 | 0.705 | 0.042 |
| | | Veg | 0.258 | 0.145 | 0.074 |
| | | BA Dead limber | 0.551 | 0.293 | 0.060 |

Fixed effect covariate coefficients, the standard error of the estimate (SE) and significance (P) are given. The variance and standard deviation (Std. Dev.) of the variance of the nested random effects are also given. In total there were 204 observations at 68 sites within 17 stands. BA = basal area.

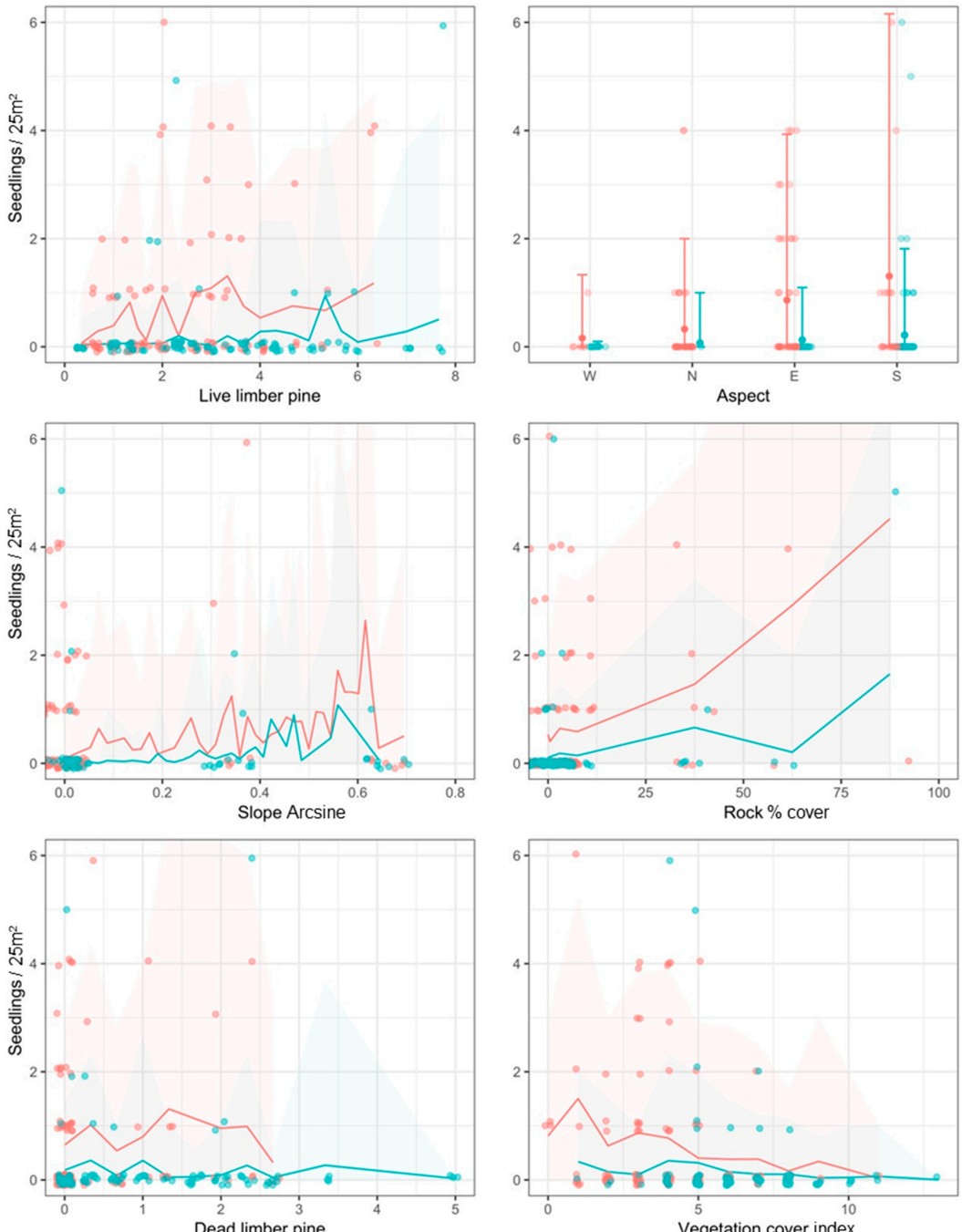

**Figure 2.** Distribution of seedling counts in northern (red) and southern (blue) ecosystems relative to: (**a**) seed availability (basal area (BA) live limber pine, m$^2$/ha), microsite, as given by (**b**) aspect (average cardinal direction), and (**c**) slope, (**d**) substrate (rock cover), (**e**) disease (BA of dead limber pine, m$_2$/ha), and (**f**) competition (vegetation cover index); variables a–d were included in the count portion of the final zero inflated Poisson model, while e–f were included in the zero inflated component of the model. Data points are slightly transparent and jittered to aid visualization. The lines represent the final model predictions conditioned on both the zero inflated and random effect components of the model and held at their average, while testing the effect of individual variables named in plots (**a**–**f**). Shaded regions represent the 95% CI of the model predictions.

### 3.2. Evaluation of Covariates in the Final Model

#### 3.2.1. Seed Availability

Site BA of live limber pine explained regeneration density better than recent seed production data (2008–2010) and was the only seed availability variable retained in the final model ($p < 0.001$, Table 5). Analyses show that seedlings increased at higher live limber pine BA on average (Table 5). Model predictions suggest that seedlings increase across a range of limber pine BA in both ecosystems, but seedling abundance is greater in the northern ecosystem across a range of seed availability. Interestingly, other seed availability covariates, namely recent cone production, and cone escape, were not retained in the final model.

#### 3.2.2. Substrate

While not abundant (<5% ground surface), the percent of rock was retained in the final model as significant explanatory variable for plots with seedlings ($p < 0.001$) (Figure 2d). Commonly available substrates, such as mineral soil and humus, which reversed availability between northern and southern ecosystems (57% versus 8%, and 9% versus 42%, mean cover respectively; Figure 3), were not retained in the global model.

#### 3.2.3. Microsite

Seedlings regenerated best on South aspects (135°–225°), particularly in comparison to West aspects (225°–315°, $p = 0.038$; Figure 2b). A variety of aspects occurred in northern ecosystem, while west aspects predominated in the southern ecosystem. Regeneration was associated with increasing slope, though not significantly ($p = 0.133$), which ranged from 5°–38°, and was retained in the final model (Table 5).

#### 3.2.4. Ecosystem

While regeneration density was 3.02 times higher in the northern ecosystem and added significant information to the final model ($p < 0.001$; Table 5), ecosystems varied in many important attributes. The northern ecosystem had lodgepole pine and white spruce instead of Douglas fir, had a wider variety of aspects (Table 1), markedly more mineral soil (Figure 3), and less understory vegetation. There was a 7.0-fold difference in density between ecosystems when the seed production:seedling ratio is computed between ecosystems (the southern ecosystem produced 1.70 times as many cones per tree × 1.35 times as many trees per hectare, divided by 0.33 times as many seedlings). This difference could be accounted for by the incremental contributions of seed, microsite, substrate covariates which enhanced regeneration, and their greater prevalence in the northern ecosystem, and lower levels of conditions that reduced regeneration (disease and competition). Even after accounting for the variety of processes found to influence regeneration in our final model, regeneration was consistently predicted to be lower in the southern ecosystem than in the northern ecosystem. These findings suggest that other processes are limiting regeneration; amongst the test covariates, we found evidence that both disease and vegetation cover can limit regeneration and may do so disproportionately in the southern ecosystem.

### 3.2.5. Disease

Site BA of dead limber pine helped explain regeneration absence and added information to our final model. A wider range of dead limber pine BA occurred in the southern ecosystem (Figure 2e). Model selection identified the negative effect of disease, by including it in the zero-inflated component of the model (Table 4). Disease appears to reduce the chances of regeneration on sites where other conditions in the model would facilitate regeneration. Intermediate host species to WPBR, such as *Ribes* spp. and *Castilleja* spp., were observed in several stands, but occurred too infrequently in regeneration plots to be included in model selection.

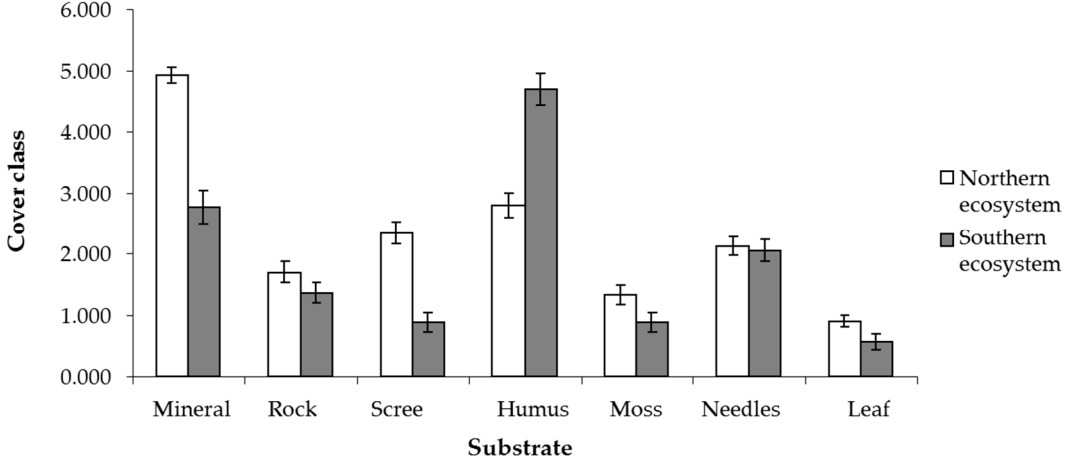

**Figure 3.** Abundance of substrates by cover classes in mature limber pine stands in northern and southern ecosystems (mean $\pm$ 1 s.e., n = 9 and 8 study stands, respectively). Percent cover classes were: 1: $\leq$ 1%, 2 = 2%–5%, 3 = 6%–10%, 4 = 11%–25%, 5 = 26%–50%, 6 = 51%–75%, and 7 = 76%–100%.

### 3.2.6. Competition

Seedling absence was associated with increasing understory cover ($p$ = 0.074; Table 5; Figure 2f). Model selection identified the negative effect of vegetation cover, by including it in the zero-inflated component of the model (Table 4). Grass was an important understory component in the southern ecosystem, while ground cover, comprised of bearberry (*Arctostaphylos uva-ursi* (L.), Spreng.), creeping juniper (*Juniperus horizontalis*, Moench), and grouseberry (*Vaccinium scoparium* Leiberg ex Coville), were common in the northern ecosystem. In the northern ecosystem, limber pine seedlings were absent in stands with abundant lodgepole pine.

### 3.3. Age Structure

Recent seedling regeneration patterns differed significantly between the northern and southern ecosystems (dmax = 15.8, $p$ = <0.02; Figure 4). The southern ecosystem has proportionately fewer young seedlings (40% < 20 years old) available to recruit into older age classes than in the north (72.8% < 20 years old). In the most recent 20 year period, regeneration appears to be continuous in both the northern and southern ecosystems (Figure 4), and there was no evidence of age distributions of seedlings differing from a uniform distribution (dmax = 6, $p$ > 0.2, and dmax = 0, $p$ > 0.5, respectively, one-sided KS test).

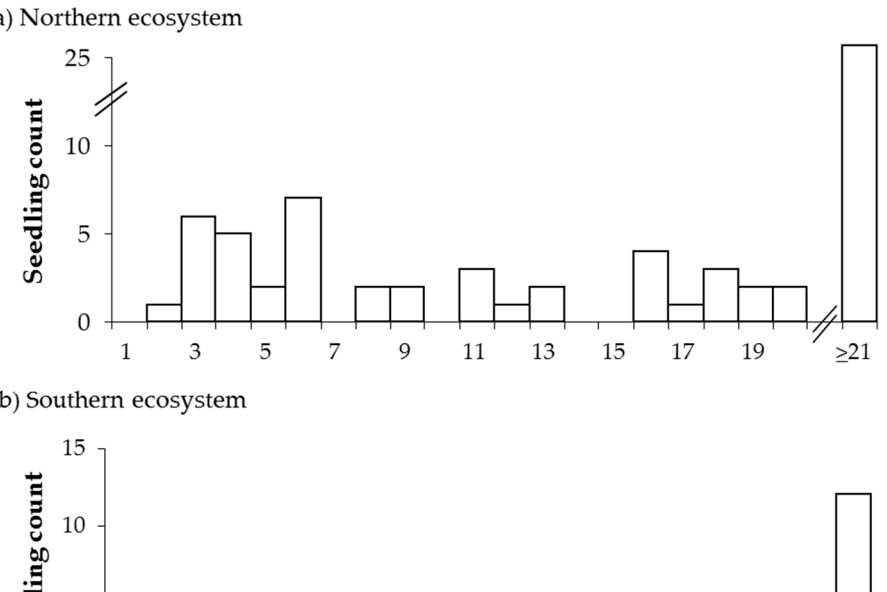

**Figure 4.** Timing of recent limber pine regeneration in regions with (**a**) low white pine blister rust infection (WPBR; n = 9 stands, 68 seedlings), and (**b**) high white pine blister rust (n = 8 stands; 20 seedlings). Ages are estimates from counts of terminal bud scar rings on seedlings, and all seedlings ≥ 21 + bud scars are grouped together due to difficulties aging older trees.

## 4. Discussion

### 4.1. Evaluation of Hypothesis Tests

It was timely to investigate the importance of regeneration processes occurring within existing stands for several reasons: (1) a rapid decline of limber pine occurred in the northern Rocky Mountain portion of its range in recent decades [17], and (2) there is a requirement for in situ maintenance of existing populations in Alberta's recovery plan [16]. We based our hypotheses on the role various biotic and abiotic covariates have on limber pine regeneration from a rich history of seed versus substrate research in many systems, and knowledge as a conservation community on both threats and challenges to the natural regeneration of limber pine. Seed availability, substrate conditions, disease, and competition feature prominently in conservation recovery plans, so a full factorial approach to testing all possible combinations of these covariates was warranted. Model selection using an information theoretic approach shows that seed, substrate, disease, and competition (final model) collectively add information to our "null" ecological model, highlighting that limber pine regeneration remains a complex process, best modelled by a variety of processes. Excluding any one of these four covariates in the factorial design, resulted in a noticeably weaker model. This was an important finding because WPBR features prominently in the literature documenting the decline of high elevation pines, and thus could have overridden other regeneration processes [7,17]. We chose to include ecosystem, and microsite in our "null model", and in all more-inclusive models in the factorial design, since there is ample evidence from plant studies that these environmental covariates affect regeneration [1], and thus provide a reasonable ecological base for modelling the regeneration process. A useful contribution of our final model is the parsing out of the influence of the covariates on the zero and the count components. Interestingly, disease and competition (vegetation cover) helped explain the zero inflated component of the model, while seed availability, site, substrate, and ecosystem are important in the

"count" component. A careful evaluation of the covariates in hypotheses is warranted to understand the regeneration processes they model, their relative contribution to the natural regeneration mechanism of limber pine, and threats facing the species.

### 4.2. Processes Facilitating Regeneration

#### 4.2.1. Seed Availability

Seed availability is a key factor controlling regeneration density in numerous studies in forest ecosystems [29,30]. Our best performing covariate in our final model, BA limber pine, measured at the site level (i.e., within 15 m of regeneration plots), was a strong predictor of regeneration density. Basal area of seed trees in wind-dispersed species and serotinous species [6,8] is a strong indicator of seed production and seedling regeneration in boreal ecosystems. Dyszoochorous species, like limber pine, which are reliant on the Clarke's nutcracker for long distance seed caching and regeneration [50], would likely have weaker relationships between BA of seed trees and seedling regeneration than other dispersal syndromes, given the reliance on the birds caching habits in determining regeneration patterns. Given the lack of documentation of barochory (gravity dispersal) as a mechanism for successful regeneration of limber pine and the high rates of post-dispersal seed predation by rodents that likely preclude germination, greater seed availability is unlikely to directly contribute to greater seedling density. We suspect that nutcrackers, which are effective dispersers of limber pine [20], responded to increasing BA of seed trees (1.7–11.2 $m^2$/ha amongst study stands), and made more caches in the same stands where seed is available, resulting in increases in regeneration density. Within this range of forest cover, mature limber pine stands clearly remain open enough to facilitate regeneration, a fact likely due to the slope and southern aspects that increase solar penetration.

One measure of actual seed available for nutcrackers, the number of cones escaping seed predators in a 2010 mast year (a proxy to historic seed availability patterns), did not provide additional information in the final model. This is not surprising, as recent cone availability may not adequately reflect the variety of species interactions that affect cone availability (i.e., predation, dispersal) that influenced the longer time span over which seedlings regenerated (>20 years). Seed production per tree varied more than 20-fold between sites, and across both ecosystems mast years were critical for limber pine cones to escape red squirrels (*Tamiasciurus hudsonicus* Erxleben, 1777) interannually [5]. Our research on squirrel predation rates in these landscapes shows that differences in cone availability are conserved even under a wide range of seed predator densities and habitat conditions left in diseased stands [5]. These findings suggest that historically more cones have been available for dispersers in the southern ecosystem, yet this has provided no comparable benefit for regeneration. One possible explanation is that there may be dispersal limitation, not seed limitation in southern sites.

High elevation pines may be disperser limited if cone production and stand health is poor [28], and cone predation by red squirrels limits seed availability in non-mast years [5]. Cone production thresholds of 1000 cone/ha for nutcracker dispersal in whitebark pine [28] suggest that BA of live trees, dead BA, and cone production would all contribute to whether limber pine is dispersal-limited. Such values of cone production were obtained in many of our study stands in the 2010 mast year (Peters, unpublished data), and likely occurred in many of the previous mast years at our study sites. Our variable dead BA of limber pine directly measures the loss of seed producing trees and may be interpreted as the reduced capacity of stands to attract avian dispersers, and a reduction in actual seed cached within seed sources. Research on the seed-dispersal behavior of nutcrackers in northern limber pine ecosystems is necessary to understand their influence on the spatial distribution of limber pine regeneration.

#### 4.2.2. Substrate Availability

Seedling age structures show that germination microsites remain available for recruitment events in mature limber pine stands, particularly in the northern ecosystem. Webster and Johnson [12]

used age structures in our study area to suggest that microsites remain available in mature stands as old as 600 years, a fact attributable to the arid environments and niche of limber pine in primary succession. Although seedlings established proportionately more often in plots with mineral soil than expected based on their relative abundance [51], particularly in the southern ecosystem, the availability of mineral soil did not add information to our final models. Coop and Schoettle [25] found that mineral soil and cobble were negatively associated with limber pine seedlings, while leaf litter was positively associated. They attributed this association to the "stability" of microsites, where litter indicates sites where plant growth could be sustained. Our modelling results identified rock, a rarely encountered substrate across both ecosystems (7% mean ground cover), as adding more information than commonly encountered substrates like mineral soil and humus, which differed markedly between ecosystems (Figure 3). Seedlings did not actually regenerate on rock, hence the variable is indicative of a terrain that may function more in nutcracker caching behavior [21], as a nurse object aiding seedling survival [24], or as a site that accumulates soil moisture near crevices, which is a determinant of limber pine regeneration [52]. While substrate helps explain seedling counts in our final model, a finer resolution assessment of substrates with respect to seedling germination and survivorship is needed to help clarify the role of substrate in the establishment phase.

### 4.3. Processes Inhibiting Regeneration

Disease-related factors appear to reduce the chances of successful limber pine regeneration at the northern limits of the species range. Several of our findings support factors identified in other studies that influenced limber pine regeneration. Firstly, regeneration was more abundant in areas with low blister rust infection and fewer WPBR-killed snags (252 and 83 individual seedlings/ha, <50 cm tall in our northern and southern ecosystems, respectively). Smith et al. [17] found nearly identical seedling cluster densities in 2009 in our study areas, with 250 and 100 seedling/ha (<1.3 m tall) in the Kootenay Plains (northern ecosystem) and Porcupine Hills/Whaleback (southern ecosystem). In range wide monitoring studies in the central and southern Rocky Mountains, seedlings <1.37 m in height averaged 141 stems/ha, but declined with WPBR infection levels (tree cankers, branch dieback, and dead BA) [53]. While earlier limber pine stand dynamics studies have suggested that age structures reflect natural stand dynamics in disease-free stands [12,25–27], our study suggests WPBR is changing regeneration patterns. We suspect that the severity and duration of WPBR infection in the southern ecosystem is contributing to limited regeneration. Currently, WPBR is increasing more rapidly at the northern limits of both limber [17] and whitebark pine's range [54] than it is in central regions, which will likely alter the age structures of regeneration in the future.

Our cone production studies indicate that in our study system, disease lowers seedling regeneration by influencing the seedling mortality process, rather than through an overriding negative effect on seed availability [5]. We have to be cautious in identifying what disease-related effects dead BA models. This variable not only reflects reduced seed availability (see Section 4.2.1), but may also contribute to zero values by reducing survivorship of seedlings. Proportionately fewer recent seedlings in the southern ecosystem may suggest that younger seedlings are succumbing to WPBR infections. Given their larger size, older seedlings may persist for several years following infection. This interpretation is hard to reconcile with other studies that have found that taller seedling have higher rates of infection and mortality than shorter seedlings in limber [17,53] and whitebark pine [54], respectively. In fact, only 0 and 1% of seedlings < 0.5 m tall were found to be infected in 2003 and 2009 range-wide monitoring surveys in Alberta [17]. We suspect that some of these differences in findings may be attributable to our attention to smaller seedling classes and new germinants that may be under-represented in health monitoring protocols, and which would not persist following infection or mortality. Provincial monitoring of WPBR in Alberta show that infection levels and mortality in adult trees increased from 1% to 20% between 2009 to 2014 (within a subset of our northern study stands), and more broadly across the northern part of limber pine range, from 2% to 11% [17]. This increase could have contributed to increased seedling mortality even in our healthier northern ecosystem,

underscoring the "resolution" challenges in inferring disease and regeneration relationships from retrospective age analyses of living seedlings. Surveys of whitebark pine regeneration at a regional level in Alberta show similar patterns to our study ecosystems, with consistent regeneration in northerly sites where WPBR infection is low, and very limited regeneration in the southern portion of the province where WPBR infection is high [54]. Collectively, these studies suggest that disease effects are relatively new in the northern ecosystem, hence many seedlings may die in coming decades, reducing their potential to sustain in situ regeneration.

### 4.4. Conservation Biology of Limber Pine

The natural regeneration processes that we identified in our analyses show that prior limber pine regeneration dynamics do not reflect current or future dynamics. Surviving seedlings in the southern ecosystem reflect historic regeneration events, culled by disease over several recent decades, while the northern ecosystem reflects continuous regeneration patterns, less affected by WPBR, a recent player on the landscape [17]. Mature limber pine stands that are disease-free are no longer present, even in the northern ecosystem; such ecosystems may still where they provide benchmarks for "expected regeneration" densities for the southern part of the species range. In mature stands, with no evidence of recent fire, range-wide surveys of seedlings vary from 172 clusters/ha in the northern Rocky Mountains (derived average from [17]) to 141 stems/ha in the central and southern Rocky Mountains [53]. Post-fire studies have reported similar densities, and while seedling regeneration is highly variable between fires, averages range from 38–314 [25], and 290–508 stems/ha [27]. As an early succession species in arid environments, "natural" rates and densities of seedling establishment are quite low, particularly compared to other conifer species that invade limber pine stands in more mesic sites [55]. Growing season moisture and soil temperature can limit first year survivorship of limber pine seedlings [52]. Climate differences between our two ecosystems may have contributed to differences in regeneration patterns; however, greater annual moisture in the southern ecosystem (525 versus 469 mm) may compensate for the added heat stress of its warmer climate ($-1.4$ vs. $-4.5$ °C, mean annual temperature). Climate differences also likely contributed to the differences in codominant tree species between ecosystems, and the ensuing difference in vegetation cover. It is important to note that both of our ecosystems support fewer seedlings than are considered necessary to maintain self-sustaining populations of limber pine in Alberta, given the threats that limber pine face provincially [16].

Age distributions of seedlings clearly show that the southern ecosystem has proportionately fewer recent seedlings available to recruit into older and larger size classes than the northern ecosystem. Depictions of age versus height suggest that seedlings take about 20 years to reach 25 cm in height (Peters, unpublished data). Difficulties with identifying terminal bud scars increase as seedlings age [56], limiting our ability to comment on recruitment dynamics after this point. Age structure differences suggest either higher rates of attrition occur in recent southern seedlings, or that barriers to regeneration in the past 20 years are disproportionately affecting the southern ecosystem. The northern ecosystem appears to reflect "healthy" limber pine regeneration dynamics, bearing a similar pattern to several studies in stands that are unaffected by WPBR, where recent seedlings are most abundant followed by saplings [12,25,26]. In contrast, the southern ecosystem was more recruitment limited over the past 20 years. While probable, we do not have the data to conclude that WPBR is directly contributing to the observed age structure, as it was beyond the scope of this study to collect seedling infection levels and relate infection to seedling survivorship. Our analyses identified that including disease in model selection improved the model fit and helped explain the absence of seedlings in heavily diseased stands. In range wide monitoring of WPBR in Alberta in 2009, seedling mortality was commonly found in infected stands [17], and may prevent successful regeneration.

One of the most intriguing and alarming findings of our study is that seed production processes do not ensure regeneration in the southern ecosystem. We found that 3.02 times more seedlings occurred in the northern ecosystem, despite only 43.5% of the net cone production observed in the southern ecosystem [4], leading to a 7-fold seed:seedling ratio difference between ecosystems. While

greater rates of limber pine cone predation by red squirrels occur in the southern ecosystem, more cones are still available in this ecosystem because the habitat supports fewer squirrels, and squirrel populations display temporal fluctuations that allow cone escape in mast years [5]. The remaining difference in seedling abundance could be accounted for by the incremental contributions of substrate and microsite covariates which enhanced regeneration, and their greater prevalence in northern ecosystems, and lower levels of conditions that reduced regeneration (disease and competition). Even after accounting for the variety of processes found to influence regeneration in our final model, regeneration was consistently predicted to be lower in the southern ecosystem than in the northern ecosystem (Figure 2). These findings suggest that other processes are limiting regeneration, which could range from differences in grazing regimes (the southern ecosystem has cattle grazing), post dispersal seed predator interactions, or other abiotic factors. Clearly, self-maintaining regeneration is possible in mature stands in the northern ecosystem, even with low seed production. While our final model indicates that regeneration will increase on average with increasing seed availability (BA limber pine), model predictions in Figure 2a suggest that greater seed production in the southern ecosystem does not appear to be adequate to compensate for a combination of disease and vegetation, or site conditions that are unfavorable in many plots in the southern ecosystem.

Given the higher seed production in our southern ecosystem, we may infer from other studies [28] that nutcrackers would be more attracted to these high cone producing regions, so the low levels of regeneration observed were unlikely to be attributable to either seed or disperser-limitation. We observed many seemingly favorable microsites that remained unoccupied, despite considerable caching activity. Further declines in seed production through WPBR induced mortality and reduced seed output in infected trees will only exacerbate in situ conservation of existing stands. Such limber pine stands may be beneficial for nutcracker assisted dispersal potential to new habitats created through fire; however, similar explorations in diseased whitebark pine systems have not led to adequate regeneration in some studies [57,58]. The relationship between fire and limber pine regeneration needs further study, particularly in northern portions of the species range, where it has only been tested in one study [59]. In our severely infected southern ecosystem, we suspect that this is a short-term recruitment opportunity with no lasting population-level benefit, given the prevalence of disease and the mortality that seedlings may experience in ensuing years. Our results suggest that when in situ conservation is prioritized, diseased stands will require greater restoration planting of WPBR-resistant seedlings because natural regeneration may not adequately restock declining stands.

## 5. Conclusions

We found support for an inclusive regeneration model that identified seed availability, substrate, microsite, and ecosystems as contributing covariates to regeneration counts, and disease and competition as covariates that helped explain the absence of regeneration. Across both ecosystems, limber pine regeneration patterns remain responsive to a variety of favorable biotic and abiotic factors. As hypothesized, our more diseased southern ecosystem had less regeneration, but there was little evidence of seed limitation caused by white pine blister rust as a contributing factor given the higher cone productivity per tree and higher seed tree basal area. Thus, despite seed availability, other biotic or environmental factors are hindering regeneration processes in our southern ecosystem. While age structure differences between ecosystems suggests that WPBR induced mortality of seedlings contributes to this finding, further study of the timing and causes of seedling mortality is needed to confirm this interpretation. Dispersal limitation may still occur if Clark's nutcrackers use southern landscapes less than the northern ecosystem; however, this is unlikely, given the much higher seed production in the southern ecosystem. Differences in regeneration may be partially attributable to suitable microsites and germination substrates being more available in the northern ecosystem, and less competing vegetation cover. Overall, seedling densities in our northern ecosystem remain comparable to studies in other parts of limber pine range, where WPBR has not been documented. Age structure of seedlings suggests that our northern populations may be self-sustaining at present, while

southern populations are not. Recent increases in WPBR at the northern limits of WPBR, combined with lower natural seed production, suggests that less natural regeneration and more WPBR-induced seedling mortality will occur in the future. This leads us to process-based recommendations for both monitoring the status of limber pine populations, and the implementation of recovery plans in jurisdictions where it is endangered. In ecosystems where natural regeneration remains responsive to a variety of biotic and abiotic factors, we recommend that attention to monitoring recent regeneration (<50 cm) is warranted, as we believe it may be a sensitive indicator of changes in the capacity of an ecosystem to sustain natural regeneration. In ecosystems where low seedling densities and the ratios of young to older seedlings suggest recruitment limitation, we recommend that recovery efforts prioritize restoration with WPBR-resistant seedlings because natural regeneration cannot be relied on to restock declining stands.

**Author Contributions:** V.S.P. conceived, designed, acquired funding, administered, and collected the data for the study. V.S.P and D.R.V. analyzed and visualized the data, and wrote the paper.

**Funding:** This research was funded by Alberta Tourism, Parks, and Recreation and Parks (Parks Division), the Alberta Conservation Association (030-00-90-161), Alberta Agriculture and Forestry (Wildfire Division, WMST FP277), and The King's University.

**Acknowledgments:** Special thanks to numerous undergraduate researchers from The King's University for faithfully counting cones, and finding seedlings, to Cyndi Smith, for providing provincial blister rust monitoring data to us, to Denyse Dawe, and several anonymous reviewers for reviewing this manuscript, and Phil DeWitt for nuanced statistical discussions.

**Conflicts of Interest:** The authors declare no conflict of interest. The funders had no role in the design of the study; in the collection, analyses, or interpretation of data; in the writing of the manuscript, or in the decision to publish the results.

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
