# Peer review of "Seed Availability Does Not Ensure Regeneration in Northern Ecosystems of the Endangered Limber Pine"

_forests, doi:10.3390/f10020146_

Reviewer 1 Report

Thank you for addressing my previous reviews. The analyses employed are much clearer in the revised text.

Author Response

No further revisions were suggested by Reviewer 1

Reviewer 2 Report

Generally, I congratulate the authors for the performed major revision. It was done very intelligently. I believe the study is better supported now and is even more convincing.

I still have one major comment (and some questions) about the variable selection/the hypothesis construction (but see my conclusion below):

Looking at Figure 2, BA of live limber pine seems to contribute actively to the zero inflation (the proportion of zeros seems to increase for lower values of BA of live limber pine). This could mean an absence of regeneration due to unavailable seeds. Vegetation cover seems to produce zero inflation also, although not so evidently.

Disease can of course explain a zero inflation, but wouldn’t you also expect that it decreases seedlings density (I mean the count component, i.e. directly in the Poisson process)? It is not clear why you did not tested BA of dead limber pine in the count component. If you tested it and it did not result significant, you should state that somewhere; if you haven’t tested, I think reader will find some incompleteness in the study… (but see my conclusion below). In the same way, vegetation cover should also be given a chance in the count component (it can modulate regeneration density more finely, than a presence/absence, 0 to 1 probability over the counts of other covariates). Note that it is not a problem for a covariate to figure simultaneously in the count and in the zero inflation component, they are doing different things (so you could keep full factorial testing).

In summary, I defend that:

i) BA of live limber pine should be part of the “null” model in the zero inflation component (and tested in the count component as it is now).

ii) BA of dead limber pine should definitely also be tested in the count component.

iii) Vegetation cover should also be tested in the count component.

Now, my conclusion on these comments/questions: I don’t feel I can impose such reasonings or analyses to the authors, because it is always possible to view things from a different perspective. I leave to the authors the decision if it is pertinent to address those questions/topics now.

Although, without addressing them, this research will be possibly viewed as less comprehensive as it could be (even if ecological research is somehow always incomplete, as we know).

Additionally, I send some minor comments and suggestions:

Page 4, lines 150: Do not delete the parenthesis of Moench as it is part of the botanical nomenclature. Correct form: “…white spruce, Picea glauca (Moench) Voss, and/or…”

Page 4, lines 150-151: In this case the correct form is: “… and/or lodgepole pine, Pinus contorta Douglas ex Loudon.” Please note the missing “ex” and Loudon not Loudan. May be abbreviated to Pinus contorta Loudon. Citing only Douglas is incorrect.

Page 4, lines 153: The correct spell is: “… Douglas fir (Pseudotsuga menziesii (Mirb.) Franco) mixtures.”. Must keep the parenthesis arround Mirb., which is an abbreviation of Mirbel, so it lacks the dot.

Page 4, line 164: “(n = 3” probably should read “n=3”. Confirm.

Page 5, line 174: “douglas fir” was first mentioned as “Douglas fir”, please standardize.

Table 1. Notes: “P. flex” should be “P. flexi”  and “P. con.” should be “P. cont.”.

Page 7, line 31: “based on these four stratum” should probably be “based on these four strata”. Confirm.

Page 7, line 42: “The units for model covariates are provided in Table 2,…” Actually, not all units are clearly given…

Page 13, line 157: Please bear in mind that when you discuss BA performance among the two different ecosystems (“particularly in the northern ecosystem; …”; etc.) you are asking for an interaction term in your model. (This is also applicable to the following subchapters.) As your models only include main effects, you can only address BA effect in general (like the average effect in both ecosystems simultaneously; note that the BA coefficient is not changing between ecosystems). All Figure 2 is based on this possible interactions between each other variable and the “ecosystem”. The figure is interesting and adds important information to the article, but careful is needed when presenting model results and discussion as they are showing different things. 

Page 14, lines 185 to 199: Great part of this text is already discussing the results. Could be moved down to Discussion. Don’t forget to refer that the small differences in climate and the fact that grazing is present in the southern ecosystem can also contribute to such differences. (I have seen after, that climate is already referred in Discusion.)

Page 15, line 201: “…helped explain regeneration density…”. I believe it would be more correct to say “…helped explain regeneration absence…” as the variable entered the model via the logistic/zero inflation component and not the Poisson/count component.

Page 15, line 203 to 206: Not very convincing. There are few plots with BA of dead limber pine greater than 2.5 m2/ha.

Page 15, line 206: m/ha should be m2/ha.

Page 15, line 209: spp. is a botanical abbreviation, should be spelled with a dot in the end, in both cases.

Page 16, line 220: Again, vegetation cover should be discussed as a zero inflation factor and not as contributing to the count/density of saplings, once it is not in the count component of the final model.

Page 16, line 221: You refer a percentage cover of 30%, but Figure 2f doesn’t seem to go that far in terms of vegetation cover. Please confirm.

Page 16, line 223: Correct spelling of botanical name: “…bearberry (Arctostaphylos uva-ursi (L.) Spreng.)…”. Just one “l” in Arctostaphylos. Keep Linnaeus abbreviated and inside the parenthesis.

Page 16, line 223: Correct spelling of botanical name: “…creeping juniper (Juniperus horizontalis Moench), and…”

Page 16, line 224: Correct spelling of botanical name: “…grouseberry (Vaccinium scoparium Leiberg ex Coville), were…”

Page 18, line 299: Which results do you refer to? Please give a reference. Nutcracker activity is not present in the current analysis so it is difficult to support such an explanation in the present work, although you can of course speculate on it, based on your expert knowledge.

Page 18, line 311: “…red squirrels (Tamiasciurus hudsonicus Erxleben, 1777)…”. Date is mandatory in animals code of nomenclature.

Page 18, line 317: Cone escape is still mentioned here, but I believe you removed it from the analysis. Please confirm.

Finally, consider citing  Peters, V.S.  2014. Nutcracker Notes 26: 12-14 in the Introduction or in the Discussion.

Author Response

Responses to Reviewer 2:

Issue 1)

We have provided evidence in table 4 that we have done what the reviewer requested, adding the 3 additional models, that additionally test BA of live limber pine in the “null” model, BA dead in the count component, and Vegetation in the count component.  These results are now reported in paragraph 2, section 3.1, but since there was no evidence to include these variables in both the count and zero component, our final model remains the best model.

We have some concerns that this “fully factorial” design may have approached a “fishing” exercise (Zurr et al. 2009); however, we are confident that our “final” model  is indeed a good model as remains the best model after testing further suggestions by the reviewer.

Minor comments:  Most are nomenclature recommendations; I deferred to the reviewers commendations given their expertise.

Page 4, 150: nomenclature updated

Page 4, 150-151: nomenclature updated

Page 4, 153: nomenclature updated

Page 4, 164: format correct as is               

Page 5 174: Standardized

Table 1 Notes:  All abbreviations standardized to 4 letters for the species name

Page 7, 31: Pluralized

Page 7, 42: Removed statement about units, as several covariates are ordinal, or factorial.  Units are provided in the methods text where appropriate.

Page 13, 157: We think this issue has mostly been addressed since we clarified language from our first draft, ensuring there is no suggestion of interactions between ecosystems.  We have now taken greater care in this sentence (we have referenced Table 5), and other places (see 461-463), to incorporate the average effect of variables in analyses with respect to the ecosystem shown in Fig. 2.   In figure 2, the caption also distinguishes how predictions are shown by fitted lines, with no suggestion that we tested for interactions.

In reviewing our results and discussion, when we’ve talked about differences between ecosystems, we have referred to specific cone averages, or density data (e.g. see 457 – 461), or age structure data (see section 4.4).  This data can be compared directly, so we feel it’s clear that there is no implication of an interaction term in analyses. 

Page 14, 185-199:  We have moved much of this text to the conservation biology section (4.4) of the discussion.  See lines 435-437 for vegetation differences, and 462-470 for a discussion of other ecological processes, including grazing.

Page 15, 201:  Good catch - Changed to absence

Page 15, 203-206:  I agree, and have removed this sentence.

Page 15, 206:  Removed with preceding sentence.

Page 15, 209:  Added “.”

Page 16, 220:  I added a sentence on zero inflation, and changed language in the paragraph to reflect this (replaced “density” with “seedling absence”)

Page 16, 221:  I removed this sentence, limiting our commentary to the vegetation index shown in fig. 2f

Page 16, 223-224:  All nomenclature corrected.   Only one of these three was misspelled, so I’m unsure why spelling was mentioned for grouseberry and juniper.

Page 18, 299: Added reference on nutcracker, and clarified the inference we are making about regeneration.

Page 18, 311: Added date

Page 18, 317:  We removed cone escape, and simplified our proposed explanation.

Citation:  Cited in the discussion, and provided a paragraph in the cover letter documenting the significant new contributions of our current publication, relative to this non-peer review format of this publication.

Page 2, 51 and throughout MS:  Formatted as suggested

Table 1:  Italics are added, and the note is expanded to include abbreviations.

Page 13, 150-151: Agreed, I have removed it from results, as it is identified adequately in our methods (Table 2). 

Please note, I have also updated the last two sentences of our conclusions to make them more reflective of our results (also updated in the abstract)

This manuscript is a resubmission of an earlier submission. The following is a list of the peer review reports and author responses from that submission.

Round  1

Reviewer 1 Report

The article is relevant, interesting and has potential for publication. Yet, some methodological issues should, in my view, be addressed before publication, namely:

Issue 1) Fitting univariate models can certainly take part of an explorative analysis, but given the bibliography on regeneration (which clearly shows that several biotic and abiotic variables can affect regeneration) this univariate hypothesis are not ecologically very sound. If concurrent hypothesis are to be tested, they should reflect combinations of variables as in the presented simplified global model.

If seed vs. substrate limitation and higher vs. lower infection are the study objective of this research (as stated in the introduction), you could analyse all models coming from a sort of full factorial design, studying all possible combinations of these 3 covariate groups (1-“seed limitation” covariates; 2-“substrate limitation” covariates and 3-“disease covariates”) testing their inclusion and exclusion in the model (2^3 = 8 possible model combinations).

However “ecosystem”, “microsite” and “competition” should, in my view, be retained in all models, as there is evidence from scientific research and it is ecologically sound that such variables affect regeneration.

Similarly, for the zero inflation component only variables expected to produce an excess of zeros should be selected (competition and disease covariates seem appropriate, but seed limitation and substrate limitation could also determine an excess of zeros; looking at the raw data could be helpful, as authors expert knowledge, but the reasoning funding the selection should be given).

Personally, as all variables are known to influence regeneration, I would prefer other approaches as inspecting all possible variable combinations, although this is very time consuming (for an example see Burnham & Anderson 2002 chapter 6.2 or Forests 2018 9(11) 694).

Issue 2) The “number of cones produced in 2010 mast year” (the “cone” covariate) corresponds to the number of cones from 10 mature trees? If so, it should not be used as is, but the density of available annual cones instead. Thus, the obtained number of cones of the 10 mature trees should be multiplied by the number of mature trees in the transect and divided by the transect area (because transects have different lengths). This should be a better variable to model sapling number, as is would be very different if a transect presents only 10 mature trees in total, or 100 mature trees.

Issue 3) Stands were characterised in 80-120 m transects. Seedlings where sampled in 25 m2 plots, while substrate and vegetation cover were sampled in 4 m2 subplots. The relationship between transects and the 25 m2 plots seem acceptable, even if a nested design could be preferable. But the relationship between the 25 m2 and the 4 m2 is more dubious (even if it is nested). Substrate and vegetation cover can vary in small patches and interact with seedlings at very fine scales. From your field experience, are the 25 m2 plots sufficiently homogeneous in terms of substrate and vegetation cover to be correctly characterised by a 4m2 subplot? I suggest that at least you discuss this in the manuscript. If the 25 m2 plots are quite heterogeneous in terms of substrate and vegetation cover, than you should not use this covariates. Old seedlings might also have germinated under different substrate conditions, but I understand that this would be too much to control for.

Issue 4) As the two ecosystems are distant and certainly differ environmentally (in terms of climate, insolation, different species interactions, other soil characteristics etc.) and given the difficulty in controlling all such factors, discussion should be quite careful and avoid excessive speculation or hasty conclusions. Recent climate change patterns are not mentioned, for example, neither possible genetic differences between sites. After reading the entire manuscript, I sense that the relevance given to the disease is greater than the evidence coming from the statistical analysis. I don’t mean that it is not relevant for limber pine conservation, but conclusions must be clearly supported by the analysis. I also feel that authors have great expertise and knowledge about the issue; that knowledge should be presented and shared in the article, but again, without mixing it with an analysis that do not support it.

Additionally, I send the following comments and suggestions:

Page 1, line 32: “the best covariates were consistent in both ecosystems”; given the statistical analysis of the manuscript, could this be otherwise? “ecosystem” is a factorial covariate in one model, thus it is not clear where you support that sentence, as you haven’t analysed ecosystems separately using different combinations of variables.

Page 1, lines 33 to 35: This conclusion is very important as it relates with the article main achievement (and title). I can’t understand the reasoning behind this sentence. It seems plausible to me to conclude that relevant variables are missing in the model… What is written as conclusion (“regeneration processes are uncoupled from seed availability”) is counterintuitive, as seed are indispensable for regeneration. I would put it as: despite seed availability, some other environmental factors are hindering regeneration (as is sometimes put in the manuscript). I am not sure what is meant with “overestimation”. Would it be possible to rephrase?

Page 2, line 51: In the whole document, I would suggest to refer the authority immediately after the Latin name, e.g.: “… endangered limber pine (Pinus flexilis James) is distinctive…”.

Page 2, line 67: Why respectively? In both northern and southern Alberta infection levels increased from 2 to 11%? Please clarify.

Page 3, lines 100 to 103. You refer to one ecosystem location. The way it is written makes you look for the second ecosystem location (which is only presented after). When you say “One ecosystem” wouldn’t “One stand” be more appropriate?

End of Introduction: Instructions for authors ask to “highlight the main conclusions” at the end of the Introduction. Please provide that.

Page 3, lines123 to 126: Even if the two selected ecosystems are in the same ecoregion, some more specific climatological information of both sites is lacking. The two areas certainly differ in some climatic variables and giving an idea of how much is that difference is relevant for the reader. Please provide some more information on this matter.

Page 3, lines 122 and 127: Please introduce the abbreviations KP and PH that you use after.

Page 4 and 5, lines 156, 157 and 159: No need to refer to the Latin name and authority again.

Table 1: Italics are missing in Latin names.

Page 7, line 4: Please define the BA (basal area) abbreviation in parentheses the first time it appears in the abstract, main text, and in figure or table captions and used consistently thereafter as asked in Instructions for authors.

Page 7, line 28: Please give more information on how this index was constructed.

Table 2 needs some additional formatting. Why did you use arcsine (not arcsin, which is an abbreviation) transformed % slope. Please, give some explanation and additional information in the text.

Page 8, line 63. Please explain how was the simplified model obtained. Was it a stepwise approach? As you are basically testing all variables and even consider that they could participate concurrently in a model (which you call global and simplified global), did you consider to test all possible combinations of variables (at least for the main effects)? See also what I have written at the end of Issue 1).

Page 9, line 76. Kolmogorov-Smirnov goodness of fit test should not be used for discrete distributions (which is the case). Some adaptations of the test do exist. Please sate if you used a test adapted to discrete distributions.

Page 9, line 100: How did you selected the covariates for the zero-inflated model component. This should also be stated in Materials and Methods.

Table 4 “k” should be in capitals, as in the footnote. Why are there so many estimated parameters in the simpler hypothesis models? It would be useful to clarify which structural parameters are involved in each model/hypothesis, in which component (count or zero-inflated), and how many non-structural parameters (inflation theta, etc.).

Figure 2, lines 123-124: “…variables in depicted in subplots…” please clarify.

Figure 3: There are no visible asterisks above bars, and Wilcoxon (not Wilcoxan) signed rank tests are not necessary here and is not applicable also (data is not paired).

Page 12, lines 131 to 133: It is not so unexpected that recent cone production (measured in 2008-2010) is not selected as covariate, as seedlings can have more than 20 years… Maybe you could reformulate. See also what I have written in Issue 2).

Page 13, lines 150 to 151: As I understand “proportion dead limber pine” is not among the set of 9 tested variables (Table 2). Should not be discussed here.

Page 14, line 181: Table 5, not 4.

Figure 4: It is stated that “…20 served as the cut-off value to distinguish between “recent seedlings” and older seedlings…” and that “Seedlings greater than 20 years were grouped into a single age category for each site…”, yet in Figure 4 it seems that you grouped the seedlings greater than 21 years. Please clarify.

Page 16, lines 238 to 240. Couldn’t greater basal area contribute to greater seed availability simply through gravity (barochory)? Additionally, as nutcrackers can cache at very long distance and prefer open sites, they could contribute also in the opposite direction of what is stated (i.e. uniformizing the geographical distribution of seeds, decreasing seed densities in places of higher basal area and increasing seed density in places of lower basal area).

Page 16, lines 243 to 254. See what I have written in Issue 2).

Page 16, lines 268 to 270. See what I have written in Issue 3).

Page 17, lines 276 to 278. A suggestion for the discussion: Rocks can also contribute to the accumulation of rainwater in crevices, increasing the soil moisture near the crevices.

Author Response:

Issue 1)
We have substantially changed the model selection procedure in response to this reviewer. We agree
that the univariate approach (univariate for hypotheses but still very much multivariate for variables)
was not the most appropriate approach. In response we have very deliberately developed an a priori
candidate model set that incorporates this reviewer’s desire for a “factorial” design. We have
accomplished this by a priori portioning the hypotheses into either the zero or count component of the
model and factorially building models with all combinations in each component. We have retained a
“null” or base set of variables in each model that are additional to the specific hypotheses we are testing
or comparing in the model selection framework. The reviewer’s suggestion from chapter 6 of Burnham
and Anderson is intriguing but given the potential to put hypotheses in either component of the model
and the number of variables associated with each hypotheses this “fully factorial” design may have
approached a “fishing” exercise. We are of the opinion that model selection is best employed for the
comparison of hypotheses that are constructed based on a biological understanding of a system and
hope the changes we have made are representative of that approach while also incorporating the good
suggestion of the reviewer. We are also pleased and confident with the “best” model we have selected,
that it is indeed a good model as it emerges out of this new model selection framework suggested by
the reviewer.
Issue 2)
We agree that the density of cones produced would be a better parameter; however, we do not have
that data. We considered constructing a combined index of cone availability (multiplying cones by basal
area, but realized there were too many untested ecological assumptions with this variable (i.e. dense
stands often have immature trees, and larger trees often have reduced cone production due to higher
infection rates). We tested the number of cones from the 10 trees as a separate measure of seed
availability since this variable was orthogonal to basal area. We have clarified our methods to show that
density of cones is accounted for by the proxy basal area (commonly used in seed production studies as
a predictor of seedling density), and that variation in transect length facilitated plot placement in stands
with a similar number of mature trees (10). Longer transects occurred when there were gaps in limber
pine tree distribution.
Issue 3)
The 4 m2 plots are nested within the 25m2 plots, and are representative of substrate conditions within
the larger plot. We have clarified in our methods that we controlled for this by orienting the larger plots
to maintain homogeneity of substrate conditions. We agree that given the age of seedlings, we could
not confirm substrate condition at the time of germination, so we did not control for this variable, nor
attempt to collect data with limited accuracy.
Issue 4)

We agree with the reviewer that we needed to be more conservative in our interpretation of disease
processes, and have now reinterpreted the analyses of disease effects in all sections. One important
aspect of this was carefully unpacking what the covariate dead basal area models from an ecological
standpoint. We have provided more explanation of the zero inflated component of our analyses, and
how it leads to our statements about disease and competition effects. We also moved a paragraph on
age structure implications from the disease section in the discussion (4.3), to conservation biology (4.4),
particularly because we acknowledge there may be several ecological processes contributing to the age
structure differences. Having supported disease effects with our model selection analyses, and not just
our age structure data, we think our analyses now back-up our connections to broader limber pine
conservation issues.
We have also provided more discussion on ecosystem differences, namely species interactions that
could alter seed availability (based on Peters et al. 2017). We have provided specific climate differences
between ecosystems, and mentioned possible effects in the discussion.
Editing comments
Page 1,32: Agreed. We have removed this statement.
Page 1,33-35: I have removed the uncoupled phrase in all but one instance in the paper, using the
suggested language of the reviewer. I have gone so far as removing “uncoupled” from the title, and find
the new title still succinctly highlights that our findings indeed warrant concern from a conservation
biology standpoint . There is a narrative that disease limits seed production, and this limits
regeneration. In contrast, we have shown, that even when there is seed, regeneration may be limited.
Page 2, 51 and throughout MS: Formatted as suggested
Page 2, 67: Deleted respectively
Page 3, 100-103: The previous sentence specifies there are two ecosystems, so I think highlighting the
uniqueness of one of these ecosystems is fine. I do mean ecosystem rather than stand since we want to
keep the emphasis on the populations, rather than one single northernmost population within this
ecosystem. Further description occurs 1 paragraph later so I don’t want to expand on the study sites in
the introduction.
End of Introduction: I’ve looked at multiple forest articles for examples and don’t see conclusions stated
in the introduction. I’ve used a similar format to other articles where the basic experimental design,
objectives, hypotheses, and implications of testing these hypotheses are stated. I’m open to further
suggestions from the editor as to meeting journal format.
Page 3, 123-126: I have provided a 10 year average of climate conditions contrasting the two
ecosystems, and referred to these differences in a comparison of the ecosystems in the discussion.
Page 3, 122, 127: Inserted abbreviations
Page 4,5, 156-159: Agreed, I removed latin names
Table 1: Italics are added, and the note is expanded to include abbreviations.
Page 7, 4: BA defined, and replaced subsequent uses of the term basal area.
Page 7, 28: Leaf area index inserted to clarify our vegetation stratum summation
Table 2: Edited table, changed to arcsine. Transforming slope, is a standard procedure since it has a
binomial distribution like proportion data. Standard methods are summarized in this table, rather than
in the text, and we prefer to keep this format.
Page 8, 63: We simplified the best model because it is known that AIC tends to overfit models (Bozdogan
1987) and since we included hypotheses in the model which may be described by multiple variables, we
inspected the significance of individual variables within each hypothesis and removed them in a
stepwise fashion. The reviewer points out that we could have constructed models based on individual
variables but the aim of our analysis was to investigate current hypotheses regarding recruitment of
limber pine. We feel that by dealing first at the level of the hypotheses and then doing a stepwise
inspection of variables within each hypothesis, this approach results in the most parsimonious model
that still adequately represented the hypotheses that were the crux of our analysis.
Page 9, 76: We have cited Zar 1996, using the test intended specifically for discrete data (see section
21.8, Zar 1996), and calculating this value by hand. We found that our stats package had previously
calculated these values using continuous data. As stated in Zar, the latter is too conservative (21.9). Our
new KS calculations have strengthened our identification of age structure differences between
ecosystems, and these are documented in section 3.3.
Page 9, 100: The reviewer is correct in noting this detail. We feel that our initial manuscript did not
adequately develop this portion of the analysis. The tricky part of the ZIP model is that both the count
and zero components must be fit simultaneously and changes in hypotheses/variables in one
component influence the significance and inclusion of hypotheses/variables in the other component.
This is not as straight forward as doing a model selection procedure for each component. However, the
reviewers comment above (ISSUE1) provided us with a framework and opportunity to better develop
the model selection procedure in this revision (for which we are thankful). We feel that our current
model selection procedure based on the construction of a priori models based on the partitioning of
hypotheses into the zero or count component of the model have allowed us to use the reviewer’s
suggested factorial approach to great effect. We hope that the changes in the descriptions of the
methods better articulate what we have now done.
Table 4: k is changed to K, thank you. The number of parameters estimated is high based on the
different structural components of the model including inflation, error and random effect terms, etc,
just as the reviewer has noted. It is also because hypotheses may contain multiple variables, so while
the model appear simple (in part because of the name it is given), it indeed remains complex. However,
from the final model presented in table 5 one can easily count off the 13 parameters that are estimated
(rows in the 3rd column). Since these are the true number of estimated entities for the model and are
used to calculate the AIC values we feel we are justified from keeping them in the table as is.
Fig 2, 123-124: Clarified language and specified plots a) – f).
Fig. 3: I deleted this sentence, as this analysis was omitted from the submitted paper.
Page 12, 131-133: I picked up on this idea in the discussion, second paragraph section 4.2.1, noting the
longer regeneration time window for seedling data, than for the cone data. As noted in the Issue 2
response, reformulating cone production was not ecologically valid based on the data we collected.
Page 13, 150-151: Agreed, I have removed it from results, as it is identified adequately in our methods
(Table 2).
Page 14, 181: Corrected number.
Fig. 4: Figure edited, with a >21 category added, and hatch marks to show discontinuity in axis for this
last age category.
Page 16, 238-240. These are important points and I have added them in the first paragraph of 4.2.1
Page 16, 243-254. My Issue 2 comment addresses this issue fully, and I also added a sentence here that
addresses the limitations of the cone availability variable directly.
Page 16, 268-270 My Issue 3 response reduces concerns about the usefulness of the substrate data. I
have been careful to use the language of “substrates seedlings are associated” with and added a
sentence at the end of 4.2.2 to show the need for finer scale evaluation of substrates.
Page 17, 276-278 I’ve added a sentence at the end of 4.2.2 about the role of crevices in enhancing soil
moisture and a reference for moisture.

Reviewer 2 Report

This paper represents an important evaluation of the factors affecting regeneration dynamics of limber pine in the context of white pine blister rust. Overall, I feel this will ultimately make a useful contribution to our understanding of recruitment dynamics within the context of a novel stressor; however, clarification is needed regarding some of the statistical approaches used before this is suitable for publication. I have outlined this concern and some minor suggestions below.

1) The authors use a model selection framework to evaluate different hypotheses regarding the factors affecting regeneration of limber pine seedlings, which is potentially a powerful approach; however, it's unclear what the relative fit of their models are relative to an uninformed, null.  In particular, model selection with AICc often includes a null or intercept-only model to serve as a general benchmark for how useful any of the candidate models are relative to an average response with associated error terms.  Right now, the authors are only comparing their candidate models to each other, so it is hard to gauge how useful these models are without the inclusion of a null.  Also, the use of the term "global model" is used throughout, but it is never defined in the text what they are referring to.  I assume this is the most complex model in the set, but this should be defined.

Minor comments:

P1, L27-29: how are you determining if this captured these processes?  Percent variance explained?

P3, L105: I would suggest replacing "hypotheses" here with "constructs" or another term

P3, L109: Again, I would suggest removing "hypotheses" here, as it about the importance of an ecological process versus hypothesis

P4, L147-148: What was the year of detection of WPBR in the northern region?

P7, L26-30: More information is needed regarding how the location of these measurements related to where seedling densities were estimated.  As it reads, these measurements did not necessarily happen in the same place, which limits ones ability to directly relate vegetation cover to regeneration.

P7, L37-39: The screening of variables based on correlations eliminates many aspects of the value of constructing a priori hypotheses to test the importance of various ecological processes on recruitment.

P8, Table 2: Looking at this table gives the impression that interactions between these different processes were not evaluated.  It appears this was in fact the case in the text, so please make sure you specify that interactions were included between different covariates.

P8, L50-66: Please include a null model in your candidate set.

P9, L106-107: What criteria were used to determine this additional complexity was not needed?

P12, L139: caption for Table 5 indicates the final model is listed in Table 2, but I could not find it there.

P12, L146: change to "humus"

P15, Figure 4. Were climate data examined to see if periods of recruitment corresponded to different levels of precipitation or snow pack?  In general, I was surprised to see little discussion of how changing climate regimes over the past several decades could also be influencing recruitment patterns.  Also, please change the x-axis labels so each year or every two years are included.  It is hard to tell what final bar represents with no labels on the last two ticks.

P15, L207-227: This section is not overly informative in terms of the actual ecological processes discovered by this research and is largely a rehash of the results section pertaining to the model selection results. I would suggest using the heading for section 4.2 "Processing facilitating regeneration" to start things off and would have an opening sentence or two that indicates you models identified several factors of importance to the regeneration process to limber pine.  Referring to these hypotheses is hard to follow and the readers are most interested in the actual phenomena you either proved or disproved.

P17, L304-306: Growing season moisture can also be quite influential on the survival of germinants and small seedlings, which again warrants some discussion of climate during this period.

Author Response:

Issue 1)
We acknowledge that having a traditional null model may be useful if we were doing a traditional
hypothesis test but our analysis is based on a model selection procedure that estimates evidence for a
model from within a candidate set. If we included a “naïve’ null (i.e. a mean number of seedlings) we
would have found it to fit terribly and any other model would be many, many AIC units better (i.e. it
would be an uninformative straw man). Because this would add little information to our analysis and
because we did not conduct a traditional hypothesis test we have chosen to not include a true naïve
null. In our revision, we did however, include what we have labelled a “null” model (but perhaps best
seen as a “base” model). This is a model informed by a limited set of variables (aspect, ecosystem,
slope) that we know may influence regeneration but were not central to the seed, substrate, and
disease hypotheses we were interested in, as such it is an informed null from which we may determine if
there is evidence that the inclusion of variables associated with the hypotheses of interest did a better
job in describing the data. We hope that this description of model selection vs. hypothesis test
adequately describes why a traditional noninformative null was not used in this analysis.
P1, L27-29: We used AIC models outlined in lines 19-20, evaluating the added information explained by
each variable (a fundamentally different approach than % variance, as in generalized linear models).
Tables 3 shows the variable assessed, and Table 5 shows the variable included in the final model, as per
screening process discussed in P7, L37-39 comment below.,
P3, L105: Substituted the term. This is a good catch as we don’t outline our A priori hypothesis
approach until later and reader will have a very specific expectation regarding our use of the term
“hypotheses” in the final paragraph of the introduction.
P3, L109: Replaced term with processes.
P4, L147-148: I added specific dates for detection in S. Alberta, and the earliest documentation in the
northern ecosystem. Unfortunately, I have not been able to find a reference for when it was actually
detected in the North.
P7, L26-30: These are important points and are the same issue raised by reviewer 1, Issue 3. Please see
my earlier response to Issue 3.
P7 L37-39: As part of a good practice of statistical model building we pre-screened variables within
hypotheses to ensure they were not overly correlated and therefore orthogonal (as advocated by Zuur
et al. 2010. “A protocol for data exploration to avoid common statistical problems” as well as in his book
cited in our manuscript). This step is important to engage in at a data exploration phase which happens
before the construction of a priori models. We feel following this protocol did not eliminate any
valuable aspects of a priori hypothesis or model construction but is part of a well-established procedure
for statistical “best practices”. It may indeed limit variables in a model but it does so for the sake of
statistical correctness.
P8 Table 2: We regret the confusion that the term “interactions” has caused. This is due, in part, to the
movement in our manuscript between the level of hypotheses in the model, which is above the level of
variables in the model. That is, variables, represent hypothesis. We checked for “crossing” at the level
of variables (i.e cones*ecosys) and did not include these complexifying interactions. However, at the
level of hypotheses, as the reviewer points out, we did include interactions in the sense of having two
hypotheses in each variable (for instance V+D), we note that this is different than the “crossing” that
could occur at the level of the hypotheses (for instance V*D) which we explicitly did not consider. We
have removed our description of interactions and have renamed the models in the model selection table
to clearly indicate that inclusion of multiple hypotheses and the nature of their “interaction” (+ vs *).
We hope that this will ameliorate any confusion and we thank the reviewer for pointing out the need for
clarification.
P8 L50-66: Please see our first comment.
P9, L106-107: Please see our comment above, P8 Table 2, regarding our use of the term interactions
between individual variables. We have omitted the statement about interactions in the results, since
they did not add information to the models.
P12, L139: Replaced with “Table 4” in the caption
P12, L146: Updated to “humus” throughout document
P15, Figure 4: I have now added some discussion commentary about the role of climate in influencing
regeneration patterns in the section 4.4, paragraph 1. Our updated age structure tests results now
reduce the value of exploring seedling recruitment episodes in the first 20 years since we now find no
significant evidence of a departure in recruitment from a uniform distribution over this period. I’ve
modified section 3.3 to reflect this.
Nonetheless, I’ve given careful thought to whether we could include a climate analysis of seedling ages,
but believe this is beyond the scope of the data collected. Having sampled seedlings non destructively,
given their provincially endangered status, we know we can’t confirm bud scar ages with ring counts,
which would also require further dendrochonological analysis for true age, given the likelihood of
missing rings owing to the slow growth, and harsh environmental conditions. This precision of age
would be required to do analyses that would invariably require correlation analyses with precipitation
for several periods in the year (winter, and germination months, combined with temperature). These
are sophisticated analyses and warrant an individual study.
We addressed labeling issues in our Figure 4 comment to Reviewer 1.
P15, L207-227 We have decided to keep this section but pitched it at a discussion level that reminds the
reader why the evaluated hypotheses were important in recovery planning, and why we chose the
factorial analysis approach proposed by reviewer 1. We also think that opening the discussion with the
hypothesis emphasis is important to retain the theoretical aspects of our study to other regeneration
studies and from there we can move to the phenomenological aspects of the specific covariates in 4.2
and 4.3. We have omitted sections that rehash the results, and ensured that these points are now made
in Section 3.1 of the results.
P17, L304-306 This is an important point, and we added some discussion of climatic differences
between ecosystems in section 4.4, at the end of paragraph 1.